# Inadequate Vitamin C Intake and Intestinal Inflammation Are Associated with Multiple Micronutrient Deficiency in Young Children: Results from a Multi-Country Birth Cohort Study

**DOI:** 10.3390/nu14071408

**Published:** 2022-03-28

**Authors:** Shah Mohammad Fahim, Md Ashraful Alam, Jinat Alam, Md Amran Gazi, Mustafa Mahfuz, Tahmeed Ahmed

**Affiliations:** 1Nutrition and Clinical Services Division, International Centre for Diarrhoeal Disease Research, Bangladesh (icddr,b), Dhaka 1212, Bangladesh; mohammad.fahim@icddrb.org (S.M.F.); mashraful@icddrb.org (M.A.A.); jinat.alam@icddrb.org (J.A.); amran.gazi@icddrb.org (M.A.G.); tahmeed@icddrb.org (T.A.); 2Faculty of Medicine and Health Technology, University of Tampere, 33100 Tampere, Finland; 3Office of the Executive Director, International Centre for Diarrhoeal Disease Research, Bangladesh (icddr,b), Dhaka 1212, Bangladesh; 4Department of Global Health, University of Washington, Seattle, WA 98195, USA; 5Department of Public Health Nutrition, James P Grant School of Public Health, BRAC University, Dhaka 1212, Bangladesh

**Keywords:** multiple micronutrient deficiencies, vitamin C, myeloperoxidase, environmental enteric dysfunction, MAL-ED study

## Abstract

Children living in resource-limited settings often suffer from multiple micronutrient deficiencies (MMD). However, there lacks evidence on the correlates of MMD in young children. We investigated the role of diets, water, sanitation and hygiene practice, enteric infections, and impaired gut health on MMD in children at 24 months of age using data from the multi-country MAL-ED birth cohort study. Co-existence of more than one micronutrient deficiency (e.g., anemia, iron, zinc, or retinol deficiency) was considered as MMD. We characterized intestinal inflammation by fecal concentrations of myeloperoxidase (MPO) and neopterin (NEO) measured in the non-diarrheal stool samples. Bayesian network analysis was applied to investigate the factors associated with MMD. A total of 1093 children were included in this analysis. Overall, 47.6% of the children had MMD, with the highest prevalence in Pakistan (90.1%) and lowest in Brazil (6.3%). MMD was inversely associated with the female sex [OR: 0.72, 95% CI: 0.54, 0.92]. A greater risk of MMD was associated with lower vitamin C intake [OR: 0.70, 95% CI: 0.48, 0.94] and increased fecal concentrations of MPO [OR: 1.31, 95% CI: 1.08, 1.51]. The study results imply the importance of effective strategies to ameliorate gut health and improve nutrient intake during the early years of life.

## 1. Introduction

Micronutrients, consisting of both vitamins and minerals, are essential for the optimal growth and development of children [1]. However, micronutrient deficiencies are highly prevalent in children living in developing countries, affecting their health and general well-being [2]. Deficiencies of certain micronutrients, for instance, iron, zinc, and vitamin A, are more common in children during their first two years of life, resulting in impaired growth, poor cognitive function, and increased risk of morbidity and mortality [3,4,5,6]. Iron deficiency is the most common form of micronutrient deficiency which contributes to anemia, as well as impaired immune and endocrine function [7,8]. According to World Health Organization (WHO), 39% of children aged below 5 years are anemic, and half of them are suffering from iron deficiency anemia [9]. On the other hand, zinc and vitamin A deficiencies are estimated to account for nearly one million child deaths per year [6,10]. Zinc deficiency leads to altered immune function and is associated with an increased incidence of enteric and respiratory infections [11]. Vitamin A is also necessary for optimum immune function and cell differentiation in the early years of life [12].

Evidence suggests that children living in resource-limited settings often suffer from multiple micronutrient deficiencies (MMD) [13,14,15,16]. Low dietary diversity, lack of adequacy, and impaired absorption of essential nutrients may play a pivotal role in the development of multiple micronutrient deficits [15]. Moreover, high rates of infection (both acute and chronic), altered intestinal health, and conditions related to improved water, sanitation, and hygiene (WASH) practice are also responsible for MMDs in low- and middle-income countries [1,15,17,18]. Recent reports demonstrated that altered gut health may mediate the relationship between nutrient intake and micronutrient status in children [1,19,20]. However, the definite role of dietary diversity, WASH behaviors, frequent exposures to multiple enteric infections, and impaired gut health on MMD at 24 months of age is yet to be explored. Therefore, we aim to conduct a secondary analysis to investigate the role of diets, water, sanitation and hygiene practice, enteric infections, and impaired gut health on MMD in children at 24 months of age using data from the MAL-ED study.

## 2. Materials and Methods

### 2.1. Study Site, Population, and Design

We have used data from a multi-country birth cohort study entitled “The Etiology, Risk Factors, and Interactions of Enteric Infections and Malnutrition and the Consequences for Child Health (MAL-ED) Study” [21]. The study was conducted in eight countries across three continents: Dhaka, Bangladesh (BGD); Bhaktapur, Nepal (NPB); Naushero Feroze, Pakistan (PKN); Vellore, India (INV); Loreto, Peru (PEL); Fortaleza, Brazil (BRF); Haydom, Tanzania (TZH); and Venda, South Africa (SAV). We have used data collected in the MAL-ED study over seven countries, excluding SAV. We excluded SAV because they did not collect data on the zinc status of the children. In all sites, data were collected longitudinally from November 2009 to October 2017. The detailed methodology of the study was published earlier [21]. Briefly, children were enrolled in the study within 17 days of birth and followed longitudinally until 24 months of age. Each site recruited children with the goal to obtain information until 24 months on more than 200 children, accounting for loss to follow-up. Exclusion criteria for enrolment in the study were maternal age of <16 years, not a singleton pregnancy, another child already enrolled in the same study, severe disease requiring hospitalization at the time of enrolment, and severe acute or chronic conditions diagnosed by a medical doctor (e.g., neonatal disease, chronic heart failure, liver disease, renal disease, cystic fibrosis, congenital conditions). The children were also excluded if they belonged to a family planning to leave the community within the next 6 months.

### 2.2. Ethics Declaration

This study is a secondary analysis using data from the MAL-ED study conducted in eight countries across three continents. The study protocol for secondary data analysis was reviewed and approved by the Institutional Review Board of the International Center for Diarrheal Disease Research, Bangladesh (icddr,b), Dhaka, Bangladesh. Moreover, each study site of the MAL-ED study obtained ethical approval from their respective institutions, and informed written consent was obtained from the parents or legal guardians of the participants [22]. 

### 2.3. Data Collection

The MAL-ED study had a standard procedure and specific criteria for data collection. Date of birth, sex, birth weight, information on initiation of breastfeeding, and anthropometry data, including length, weight, and head circumference, were recorded at enrolment. Household and socio-economic status (SES) data were collected at 6, 12, 18, and 24 months of age. Data on morbidity and illnesses were reported by the mothers or caregivers regularly from birth to 24 months of age. Trained field staff visited the households bi-weekly and asked the caregivers about breastfeeding and non-breast milk food consumption of the children. Mothers or caregivers were also interviewed monthly to obtain dietary data on breastfeeding, as well as the introduction of complementary foods. From 9 months of age, 24-h dietary recall data were collected monthly from the caregivers of each child.

### 2.4. Biological Sample Collection

Biological samples, including blood and stool, were collected at regular intervals from each child. Blood samples were collected at 7, 15, and 24 months of age. However, micronutrient status was measured in the blood samples collected at 24 months of age. Non-diarrheal stool samples were collected during monthly home visits, and diarrheal stool samples were collected during the diarrheal episodes.

### 2.5. Laboratory Analysis

In the MAL-ED study, stool samples were analyzed to detect the presence of enteric pathogens using microscopy, culture, enzyme-linked immunosorbent assay (ELISA), and polymerase chain reaction (PCR), as appropriate. Fecal biomarkers of intestinal inflammation and altered gut permeability, for instance, myeloperoxidase (MPO), neopterin (NEO), and alpha-1-antitrypsin (AAT), were measured in stool samples by ELISA. Blood samples were analyzed to evaluate the micronutrient status, such as iron (plasma ferritin and transferrin receptor (TfR)), zinc (plasma zinc), and vitamin A (plasma retinol). Hemoglobin concentration was determined using the HemoCue method. Since acute phase response to infections may alter the micronutrients status, the plasma level of α-1-acid glycoprotein (AGP) was also measured using ELISA in this study.

### 2.6. Variables Used in This Analysis

Outcome variable: The outcome variable for this analysis was multiple micronutrient deficiencies (MMD). Co-existence of more than one micronutrient deficiency (e.g., anemia, iron, zinc, or retinol deficiency) was considered as MMD. In this analysis, iron deficiency was defined as a plasma ferritin concentration < 12.0 μg/L and TfR concentration > 8.3 mg/L, zinc deficiency as a zinc concentration < 9.9 μmol/L, and retinol (vitamin A) deficiency as a retinol concentration < 0.70 μmol/L. In the MAL-ED study, hemoglobin concentration was adjusted for altitude where appropriate (NEB and TZH), and anemia was identified by hemoglobin < 11.0 g/dL.

Exposure variables: Age, sex, socio-demographic variables, dietary intake, WASH variables, enteropathogen exposures, gut biomarkers, biomarkers of systemic inflammation, and morbidity were assessed as the exposure variables in this analysis. The concentration of the plasma and stool biomarkers was log-transformed, and, subsequently, the mean was calculated over observations from 9 to 24 months. Dietary intake was calculated from the 24-h food recall data. The average intake from non-breast milk foods was calculated from 9 to 24 months of age. Breastfeeding was characterized by the percentage of days a child was breastfed between 9 and 24 months. We have calculated a new variable “pathogen score” to estimate the burden of all the enteropathogens tested in the study. The score was calculated using the average number of enteropathogens detected in non-diarrheal stool divided by the total number of stool samples tested for each participant [22].

### 2.7. Statistical Analysis

At first, overall and country-wise, general characteristics of the children were described using frequencies with percentages for categorical variables and mean with standard deviation for symmetric numeric variables. The asymmetric numeric variables were reported with median and inter-quartile range (IQR). Generalized Linear Mixed Effects Models (GLMM) were used to estimate the associations between exposure variables and the outcome of interest. This is a multi-country dataset that contains clusters of non-independent observational units namely ‘country’. This is assumed that measurements within a country might be more analogous than measurements from different countries. Hence, we used GLMM to adjust this clustering effect, considering the intercept of the variable ‘country’ as a random intercept. This approach allowed a robust estimation of variance in the outcome variable within and between the clusters.

Associations were estimated by unadjusted and adjusted odds ratios with 95% confidence intervals obtained from bivariate and multivariable analyses, respectively. Variables with *p*-values less than 0.20 in bivariate analyses were included in the multivariable analysis. Prior to constructing a multivariable model, the correlations (Spearman’s rank, ρ) between candidate variables were examined to check the collinearity. If the variables were found to be correlated (e.g., ρ > 0.4), biological justification and evidence from prior studies were considered to retain the variables in the multivariable model. Multicollinearity was also checked with variance inflation factor (VIF).

Based on the conceptual framework stated in Figure 1, a Bayesian network was developed to document the relationship between dietary adequacy, dietary diversity, WASH variables, enteric pathogens, intestinal inflammation, gut permeability, and systemic inflammation, and their associations with MMD [1,23]. Bayesian network analysis is a rigorous statistical model that allows the evaluation of statistical associations amongst variables, including one or more outcome measures. We performed sensitivity analyses to evaluate the impact of variables included in the Bayesian network and confirm the findings for generalizability. A complete case analysis was applied for all the analyses, and statistical significance was determined with a two-sided *p*-value less than 0.05. All the statistical analyses were performed using R version 4.0.5 (https://www.r-project.org, accessed on 31 December 2021, Foundation for Statistical Computing, Vienna, Austria) software. We used the lme4 package for GLMM analysis. The Bayesian network was constructed using JAGS 4.2.0 (http://mcmc-jags.sourceforge.net, last accessed on 31 December 2021) [24].

## 3. Results

### 3.1. Basic Socio-Demographic Characteristics

At 24 months, 1294 (71%) of the 1831 children enrolled throughout the 7 sites had blood drawn. Due to missing data for micronutrient variables in 201 children, the final analytic sample size was 1093 children (Table 1).

The descriptive characteristics of the participants are described in Table 2 and Appendix A. The average birth weights ranged from 1.52 kg in PKN to 3.36 kg in BRF, and 48% of children were female. Overall, median energy intake from complementary foods (9 to 24 months) was 699 Kcal/d with the lowest median energy intake in BGD (338 Kcal/d) and the highest median energy intake in TZH (994 Kcal/d). The 7 locations had different median iron intakes, ranging from 1.31 mg/d in NPB to 14.2 mg/d in BRF. The zinc intakes in the 7 areas varied, ranging from 1.13 mg/d in BGD to 9.05 mg/d in BRF. The median vitamin C intake differed among the 7 sites, ranging from 5.84 mg/d in TZH to 101 mg/d in BRF. The geometric mean of MPO (birth to 24 months) ranged from 2515 ng/mL in BRF to 7044 ng/mL in INV.

### 3.2. Prevalence of Multiple Micronutrient Deficiencies

Overall, 47.6% of the children had MMD with the highest prevalence in Pakistan (90.1%) and lowest in Brazil (6.3%). The prevalence of MMD by study sites are shown in Figure 2.

### 3.3. Factors Associated with Multiple Micronutrient Deficiencies

Figure 3 depicts the Bayesian network’s results, with colors showing the direction of the relationship (red, positive; blue, negative). The numerical values of the associations are reported in Appendix A. MMD was inversely associated with female sex [OR: 0.72, 95% CI: 0.54, 0.92]. A greater risk of MMD was associated with lower vitamin C intake [OR: 0.70, 95% CI: 0.48, 0.94] and increased fecal concentrations of MPO [OR: 1.31, 95% CI: 1.08, 1.51], the biomarker of intestinal inflammation (Appendix A and Figure 3).

MPO was significantly associated with protein intake [β: −0.47; 95% CI: −0.69, −0.25], bacterial infection [β: 0.08; 95% CI: 0.02, 0.14], and diarrheal incidence [β: −0.17; 95% CI: −0.24, −0.11]. Vitamin-C intake and bacterial load were negatively associated with households with high crowding index (Appendix A and Figure 3).

Viral load was inversely associated with monthly family income, while parasite score was significantly associated with maternal schooling years. Zinc, thiamine, and niacin intakes were positively associated with monthly family income, and the relationships were statistically significant. Likewise, protein, iron, zinc, niacin, riboflavin, and vitamin A intakes were significantly associated with maternal education years (Appendix A and Figure 3).

## 4. Discussion

This is a secondary analysis of data obtained from a previously conducted, multi-country birth cohort study, which assessed the etiology, risk factors, and interactions of enteric infections and malnutrition and the consequences for child health [21]. Our study results demonstrated that almost half of the children were suffering from MMD across the study sites. Additionally, female sex and vitamin C intake were found to be protective against MMD, while children with MMD were more likely to have increased fecal concentrations of MPO measured in their stool samples. 

We found a statistically significant negative association between MMD and female sex. There is evidence that micronutrient deficiencies are affected by sex, but this is often culturally specific where sex discrimination is common [25]. Besides, micronutrient deficiency may also be affected by sociodemographic status, environmental context, and dietary habits [25,26,27]. McLaren DS et al. found that vitamin A deficiency can be 10 times greater in male children than their female counterparts [28]. A recent study also showed that female participants responded better to oral nutritional supplements given for a duration of six months. They had a better improvement in hemoglobin, albumin, and zinc status compared to the male subjects [29]. However, no significant difference was found between boys and girls of 1 to 8 years of age in terms of hemoglobin, iron, and transferrin levels in a study conducted in Thailand, although they reported significantly higher concentrations of ferritin among girls [30]. Studies conducted in Haiti and Indonesia also reported that boys had a higher prevalence of anemia compared to girls [31,32]. All these reports support our finding on the inverse relationship between MMD and female sex [28,33].

We observed that a lower vitamin C intake was associated with a greater risk of MMD. Moreover, a higher proportion of MMD was reported in countries with low vitamin C intake. In most countries, the intake of vitamin C was less than the recommended daily amount for children of this age. Although the prevalence of MMD was found lowest in BRF, the vitamin C intake was much higher than the desirable daily dose among the children enrolled in BRF. Vitamin C, commonly known as ascorbic acid, is an essential water-soluble micronutrient for humans. It is recommended to obtain through diets in an adequate amount to prevent the deficiency and further adverse consequences. There is sufficient evidence that vitamin C improves immune functions and prevents infections in children [34]. Earlier evidence suggests that vitamin C helps to enhance the absorption of inorganic iron, which plays important role in the metabolism of folic acid, as well as some amino acids and hormones [35]. Vitamin C has been shown to increase iron absorption by capturing non-heme iron and storing it in a form that can be more easily absorbed in our body [36,37,38]. Vitamin C stimulates iron absorption using two mechanisms: by reducing ferric to ferrous iron (form required for uptake into mucosal cells), and the prevention of the formulation of insoluble and unabsorbable iron compounds [36,38]. The previous report indicated that children who consumed a plant-based diet with low vitamin C suffered more from iron deficiency anemia that can be improved through supplementation of vitamin C [39]. Another study conducted in the adult population showed that ascorbic acid supplementation for two months improved iron status among strict vegetarian adults [40]. Therefore, it is imperative to ensure vitamin C adequacy in diets in order to prevent MMD in children. Our finding also implies the importance of adequate vitamin C intake to avert multiple micronutrient deficits during the early years of life. To that end, the impact of vitamin C intake should be considered while developing strategies to improve micronutrient status among children living in developing countries [36].

MPO is a biomarker of intestinal inflammation, as well as environmental enteric dysfunction (EED) [41]. EED is an asymptomatic small intestinal ailment highly prevalent among children living in tropical climates [42,43]. The condition is characterized by intestinal inflammation and loss of gut integrity [42]. MPO is a product of neutrophils, which is liberated in response to intestinal mucosal injury [44]. MPO was found to be associated with anemia [1]. Moreover, the increased fecal concentration of MPO indicates direct iron loss from the body, as MPO is an iron-holding protein [1]. It is also worth mentioning that MMD and infectious diseases aggravate one another like a circular argument and cause micronutrient deficiency through upregulation of neutrophils [45]. As mentioned earlier, our study identified that increased fecal concentration of MPO is associated with a greater risk of MMD. This finding suggests that, apart from poor dietary intake, intestinal inflammation, as well as EED, may play a potential role in the development of MMD during early childhood [46].

We have used data from a multi-country birth cohort study that ascertains the generalizability of the findings among children under two living in different geographic locations. Moreover, we followed advanced statistical methods and applied the Bayesian approach to identify the factors associated with MMD in children. This is a strength of this analysis. However, recall bias for dietary data and reporting bias for data on WASH practices are the limitations of this study. Vitamin C is a good indicator of fruit and vegetable intake. Low vitamin C intake indicates a lower intake of fresh fruit and vegetable. Hence, it would be better if we could look at the foods that are missing from the diets, particularly fruits and vegetables. However, the lack of adequate data on fresh fruit and vegetable intake from each site restricts us to include this information in this manuscript. It is another limitation of this study.

## 5. Conclusions

This study reports that nearly half of the children were suffering from MMD at 24 months of age. The study findings provide evidence on the relationship between inflammatory biomarkers of EED and MMD among young children living in multiple resource-limited settings. It also demonstrates that increased intake of vitamin C from complementary foods may reduce the risk of MMD. This finding emphasizes the necessity of increasing fresh fruit and vegetable intake during the early years of life. The results also imply the importance of effective strategies to prevent and control EED and improve nutrient intake in order to reduce MMD during the early years of life.

## Figures and Tables

**Figure 1 nutrients-14-01408-f001:**
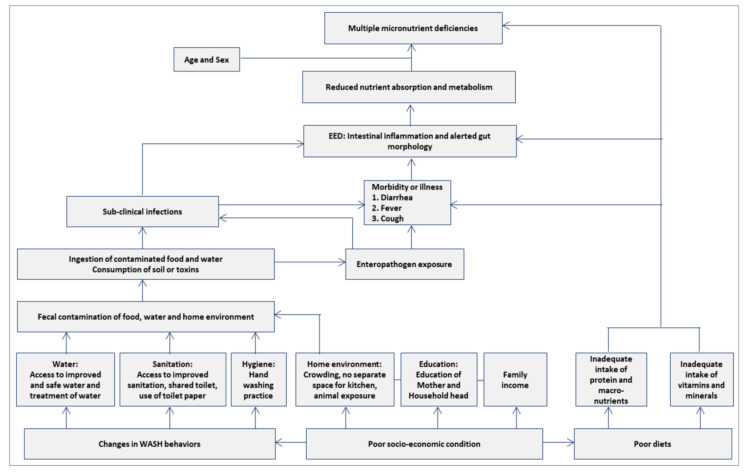
Conceptual framework.

**Figure 2 nutrients-14-01408-f002:**
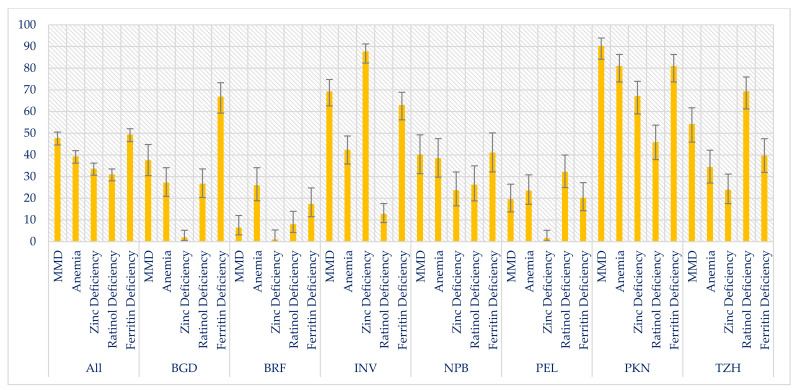
Prevalence of multiple micronutrient deficiencies, overall and by MAL-ED study sites.

**Figure 3 nutrients-14-01408-f003:**
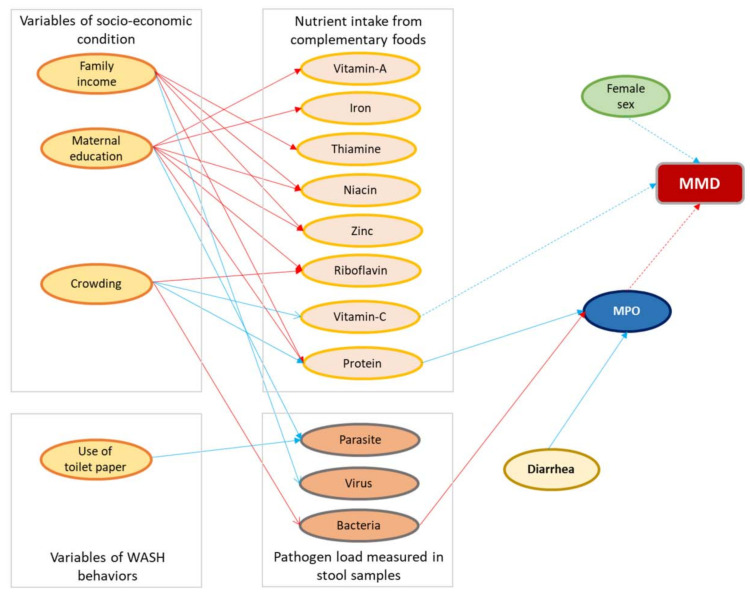
Results of the multivariate Bayesian network. Arcs (arrows) are shown for parameters that did not include zero in the 95th percentile credibility interval. Positive associations are shown in red and negative in blue. The arcs with solid lines indicate linear associations and dashed lines indicate log odds. MPO, myeloperoxidase; MMD, multiple micronutrient deficiency.

**Table 1 nutrients-14-01408-t001:** Cohort profile by site. BGD, Bangladesh; INV, India; NPB, Nepal; PKN, Pakistan; BRF, Brazil; PEL, Peru; TZH, Tanzania.

	BGD	INV	NPB	PKN	BRF	PEL	TZH	ALL
Enrolled	265	251	240	277	233	303	262	1831
Blood draw at 24 months	177	225	116	222	143	195	216	1294
Missing hemoglobin	1	0	0	0	0	0	8	9
Missing zinc	2	1	1	49	15	44	54	166
Missing retinol	0	0	0	19	0	0	1	20
Missing ferritin	0	1	0	3	0	1	1	6
Analytic total	174	223	115	151	128	150	152	1093

**Table 2 nutrients-14-01408-t002:** Descriptive characteristics of the study participants, overall and by MAL-ED study sites.

	BGD	BRF	INV	NPB	PEL	PKN	TZH	Overall
	(*n* = 174)	(*n* = 128)	(*n* = 223)	(*n* = 115)	(*n* = 150)	(*n* = 151)	(*n* = 152)	(*n* = 1093)
**Female child, *n* (%)**	86 (49.4%)	58 (45.3%)	120 (53.8%)	51 (44.3%)	62 (41.3%)	73 (48.3%)	73 (48.0%)	523 (47.9%)
Birth weight (kg), Mean ± SD	2.81 ± 0.41	3.36 ± 0.52	2.89 ± 0.44	2.95 ± 0.40	3.13 ± 0.42	1.52 ± 1.46	3.27 ± 0.49	2.83 ± 0.88
Daily energy (Kcal) intake, Median [IQR]	338 [272, 412]	987 [824, 1140]	739 [600, 894]	392 [328, 481]	728 [610, 876]	619 [470, 782]	994 [888, 1130]	699 [452, 917]
Daily protein (gm) intake, Median [IQR]	9.37 [7.83, 12.0]	40.3 [34.3, 48.7]	20.7 [16.4, 26.8]	10.6 [8.09, 13.7]	19.5 [16.3, 22.6]	15.8 [11.9, 20.4]	28.2 [24.6, 34.0]	19.2 [12.2, 27.5]
Daily carbohydrates (gm) intake, Median [IQR]	56.1 [45.0, 67.9]	138 [113, 157]	112 [93.6, 133]	58.8 [49.4, 69.5]	126 [109, 154]	86.8 [69.6, 107]	185 [163, 207]	107 [70.0, 143]
Daily iron (mg) intake, Median [IQR]	1.40 [1.09, 1.78]	14.2 [11.3, 17.4]	2.38 [2.05, 2.95]	1.31 [1.06, 1.70]	3.77 [2.99, 4.80]	2.09 [1.69, 2.78]	8.31 [7.23, 9.14]	2.70 [1.72, 6.90]
Daily folate (ug) intake, Median [IQR]	34.3 [28.8, 44.8]	166 [133, 201]	82.0 [68.5, 101]	40.3 [32.8, 53.9]	57.8 [43.0, 71.7]	58.7 [47.0, 78.3]	65.0 [56.2, 77.3]	63.6 [44.2, 88.8]
Daily zinc (mg) intake, Median [IQR]	1.13 [0.90, 1.42]	9.05 [7.27, 11.0]	2.85 [2.24, 3.69]	1.28 [1.00, 1.71]	2.20 [1.87, 2.74]	1.94 [1.46, 2.54]	5.25 [4.55, 6.03]	2.43 [1.48, 4.54]
Daily thiamin (mg) intake, Median [IQR]	0.18 [0.15, 0.23]	0.77 [0.65, 0.95]	0.25 [0.20, 0.32]	0.16 [0.12, 0.22]	0.28 [0.23, 0.37]	0.41 [0.33, 0.52]	0.56 [0.47, 0.63]	0.30 [0.20, 0.52]
Daily niacin (mg) intake, Median [IQR]	2.64 [2.08, 3.19]	9.47 [7.51, 11.3]	2.66 [2.12, 3.22]	1.60 [1.25, 2.07]	3.96 [3.13, 4.60]	4.37 [3.43, 5.61]	5.44 [4.63, 6.12]	3.53 [2.44, 5.35]
Daily riboflavin (mg) intake, Median [IQR]	0.20 [0.15, 0.30]	1.52 [1.27, 1.87]	0.44 [0.29, 0.79]	0.25 [0.17, 0.37]	0.47 [0.35, 0.65]	0.49 [0.36, 0.74]	0.97 [0.64, 1.34]	0.47 [0.28, 0.94]
Daily Vitamin-A (ug) intake, Median [IQR]	52.8 [35.9, 77.4]	955 [754, 1150]	181 [116, 284]	83.9 [61.0, 114]	219 [148, 335]	150 [93.0, 220]	136 [91.5, 184]	147 [81.2, 271]
Daily Vitamin-C (mg) intake, Median [IQR]	10.1 [6.88, 16.2]	101 [77.2, 121]	9.67 [7.92, 11.9]	7.28 [5.08, 10.0]	55.4 [28.5, 114]	11.6 [7.38, 17.9]	5.84 [3.96, 7.89]	11.2 [7.11, 27.5]
Daily Vitamin-B6 (mg) intake, Median [IQR]	0.26 [0.20, 0.32]	1.03 [0.79, 1.51]	0.27 [0.21, 0.34]	0.23 [0.19, 0.29]	0.37 [0.30, 0.45]	0.42 [0.32, 0.56]	0.15 [0.11, 0.20]	0.30 [0.21, 0.44]
Daily Vitamin-B12 (ug) intake, Median [IQR]	0.37 [0.24, 0.60]	4.61 [3.96, 5.59]	0.62 [0.39, 0.96]	0.51 [0.31, 0.84]	0.94 [0.68, 1.29]	0.68 [0.42, 1.21]	1.18 [0.72, 1.75]	0.77 [0.42, 1.44]
EBF duration in days, Median [IQR]	109 [59.3, 157]	56.5 [27.0, 99.3]	79.0 [39.5, 111]	43.0 [17.5, 96.0]	14.5 [6.00, 56.8]	10.0 [5.00, 14.0]	34.0 [19.8, 61.0]	42.0 [14.0, 95.0]
Myeloperoxidase (ng/mL) in log scale, Mean ± SD	8.31 ± 0.41	7.83 ± 0.73	8.86 ± 0.46	8.55 ± 0.47	8.83 ± 0.48	8.31 ± 0.55	8.53 ± 0.41	8.49 ± 0.60
Neopterin (nmol/L) in log scale, Mean ± SD	6.87 ± 0.45	7.28 ± 0.38	7.41 ± 0.37	7.47 ± 0.28	7.72 ± 0.37	6.14 ± 0.39	6.62 ± 0.51	7.07 ± 0.65
α-1 antitrypsin (mg/g) in log scale, Mean ± SD	−0.95 ± 0.37	−1.30 ± 0.49	−1.11 ± 0.37	−0.76 ± 0.37	−0.98 ± 0.42	−1.78 ± 0.58	−1.46 ± 0.48	−1.19 ± 0.54
Hemoglobin (g/dL), Median [IQR]	11.9 [10.9, 12.7]	12.0 [10.6, 13.3]	11.2 [10.2, 11.9]	11.3 [10.6, 11.7]	11.7 [11.0, 12.5]	9.80 [8.60, 10.7]	11.4 [10.5, 12.2]	11.3 [10.2, 12.2]
Zinc (μmol/L), Median [IQR]	11.8 [10.9, 13.2]	13.6 [12.9, 15.4]	8.50 [8.00, 9.20]	11.6 [10.1, 13.0]	14.2 [11.8, 16.7]	8.00 [6.00, 10.6]	11.2 [10.0, 12.7]	11.1 [9.00, 13.3]
Ratinol (μmol/L), Median [IQR]	24.0 [19.9, 28.0]	30.7 [25.3, 35.8]	29.2 [23.2, 36.1]	24.5 [19.9, 30.4]	22.8 [18.9, 28.9]	20.6 [16.5, 25.0]	17.2 [13.8, 20.7]	24.1 [18.9, 30.5]
Ferritin (μg/L), Median [IQR]	7.75 [4.70, 14.2]	25.3 [15.0, 36.4]	8.60 [4.35, 19.7]	14.8 [7.10, 24.2]	25.9 [12.8, 37.3]	4.00 [2.00, 8.10]	15.0 [7.70, 24.5]	12.3 [5.30, 25.7]
Bacteria load, Mean ± SD	0.91 ± 0.25	1.05 ± 0.33	0.79 ± 0.28	0.70 ± 0.22	0.56 ± 0.24	1.14 ± 0.31	1.04 ± 0.31	0.88 ± 0.34
Parasite load, Mean ± SD	0.12 ± 0.10	0.14 ± 0.20	0.21 ± 0.19	0.14 ± 0.16	0.32 ± 0.21	0.45 ± 0.23	0.23 ± 0.18	0.23 ± 0.21
Virus load, Mean ± SD	0.08 ± 0.08	0.03 ± 0.06	0.06 ± 0.06	0.06 ± 0.07	0.08 ± 0.07	0.07 ± 0.08	0.09 ± 0.09	0.07 ± 0.08

## Data Availability

All relevant data are within the manuscript.

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
