# Peer review of "Inadequate Vitamin C Intake and Intestinal Inflammation Are Associated with Multiple Micronutrient Deficiency in Young Children: Results from a Multi-Country Birth Cohort Study"

_nutrients, 2022, doi:10.3390/nu14071408_

Round 1

Reviewer 1 Report

The article titled: “Inadequate Vitamin C Intake and Intestinal Inflammation Are Associated with Multiple Micronutrient Deficiency in Young Children: Results from a Multi-Country Birth Cohort Study” shows that a significant percentage of children living in prenatal settings with limited resources had MMD at 24 months of age.

The article reviews data obtained from the MAL-ED study to investigate the role of diets, water, sanitation, hygiene practices, enteric infections, and impaired gut health in MMD in 24-month-olds. .

Similar to what was published by McCormick et al., (2019), intestinal inflammation is associated with MMD. What is new in this article is that the authors focused on vitamin C intake and demonstrated a negative correlation with MMD. They also discussed the importance of improving the intake of this vitamin to reduce MMD during the first years of life.

The authors applied the Bayesian approach to identify factors associated with MMD in children. Although Table 2 shows a large amount of data and is too long to cover the median effects of each predictor on each response variable within the model; a summary of these associations is presented in figure 4 showing a positive or negative relationship with different colors. The article's conclusion is based on this analysis and shows that an increased risk of MMD was associated with lower vitamin C intake and higher fecal concentrations of MPO, the biomarker of intestinal inflammation. In addition, vitamin C intake and bacterial load were negatively associated with households with a high rate of overcrowding.

Specific comments

The authors wrote that "median vitamin C intake differed among the seven sites, from 5.84 mg/d in BGD to 101 mg/d in BRF." They also showed that overall 47.6% of children had MMD, with the highest prevalence in Pakistan (90.1%) and the lowest in Brazil (6.3%). Taking into account the importance shown for the intake of this vitamin in the conclusion, the authors could discuss about these values, especially for BRF, are the amounts in the desirable range for children of this age?

Table 2 is very extent, one possibility to improve it could be that the information can be separated in a similar way to how it was grouped in figure 4, in one table showing the responses related to the intake of nutrients from complementary foods and in another the load of pathogens measured in stool samples.

Reviewer 2 Report

At face value, this manuscript positions itself to be a paper analyzing the nutritional needs of children and their relationship to inflammation. While the study (understandably) has collected a great deal of information to examine this relationship, the manuscript suffers from the presentation of that information. In short, it is too dense and convoluted to understand the nutrition story that is trying to be conveyed.

Overall, it is recommended that this paper undergo extensive revision to limit the data presented to nutrition information and the specifics surrounding micronutrient deficiencies. 

General comments follow:

  1. Figure 2 is not necessary to provide as a figure. It would be more efficient to present it as a table.
  2. Table 1 is massive and should be concentrated down to relevant information. Nutrient intakes and/or makers of inflammation are the only pieces of information that seem relevant - although other variables of interest can be kept. The remaining data should be put in a supplemental data file.
  3. Where are the results of the blood analysis? Why are they not included in this manuscript? If these are the information used to determine the data in Figure 3, then the blood data results should be presented as individual panels, with the MMD (the synthesis of that data) presented as an additional panel. While every blood measurement does not have to be represented, some choice markers should be displayed.
  4. It is unclear what data from Table 2 is relevant. The table is large and left to explain itself rather than incorporated into a narrative within the text of the manuscript. If nothing else, there needs to be a more efficient way to represent the data, if not indicating what data is more important/relevant than others.
  5. In the conclusions of the manuscript, the authors talk about vitamin C intake as if it occurred in a vacuum, instead of looking at the food items that are involved in delivering vitamin C to these individuals. In fact, vitamin C is typically a good indicator of fruit and vegetable intake and it would follow that fresh fruit and vegetable intake is low in individuals with low vitamin C intake. Therefore, it would make more sense to look at the foods that are missing from the diets that may contribute to MMD (rather than the individual micronutrients). 

Round 2

Reviewer 2 Report

The manuscript is improved by the authors' changes, with the exception of style, formatting, grammar, and spelling errors.